# Story Generation with Commonsense Knowledge Graphs and Axioms

**Filip Ilievski**                                    ILIEVSKI@ISI.EDU
**Jay Pujara**                                        JPUJARA@ISI.EDU
**Hanzhi Zhang**                                      HANZHIZH@ISI.EDU
*Information Sciences Institute, University of Southern California*
*Marina del Rey, CA, USA*

## Abstract

Humans can understand stories, and the rich interactions between agents, locations, and events, seamlessly. However, state-of-the-art reasoning models struggle with understanding, completing, or explaining stories, often due to the complexity of the underlying common sense necessary for comprehension. One potential reason models perform poorly is the lack of large-scale training data that supplies annotations of the common sense necessary for story understanding. In this paper, we investigate the generation of stories at scale, by combining commonsense axioms with commonsense knowledge graphs to produce stories annotated with common sense. We first demonstrate that commonsense axioms and commonsense knowledge graphs are sufficient to capture the underlying narratives in a popular story corpus. Our method aligns story types with commonsense axioms, and queries to a commonsense knowledge graph, enabling the generation of hundreds of thousands of stories. We evaluate these stories for sensibility and interestingness through a crowdsourcing task. Using our story corpus, we also design a probing task with questions for three exemplar story types. Our results show that our method generates story endings of a higher quality compared to the current generative language models. This work points to key open challenges around generating better stories, providing more comprehensive explanations, and building models that can explain any story with axioms.

## 1. Introduction

Humans can understand stories, and the rich interactions between agents, locations, and events, seamlessly. Being able to fill in the blanks with background, commonsense (CS) knowledge about agents and events is required for understanding and explaining stories [Charniak, 1972, Schank and Abelson, 1975], in accordance with pragmatic principles of non-redundancy of information and avoiding to state the obvious [Grice, 1975]. For example, in order to understand that forming a new friendship causes a person to be happy, we need to understand both the key axioms about human psychology and goals (e.g., happiness is a consequence of achieving a goal), as well as universal human knowledge (e.g., people generally have an intrinsic goal to make friends).

State-of-the-art AI reasoning models struggle with understanding, completing, or explaining stories, often due to the complexity of the underlying common sense necessary for comprehension. Neural (language) models could be tuned to understand or explain stories [Narang et al., 2020], but this is hindered by the lack of large-scale training data that supplies annotations of the common sense necessary for story understanding.

Creating large-scale data for training neural models is an open challenge. High-quality story understanding and completion tasks [Mostafazadeh et al., 2017, Urbanek et al., 2019] can be created by crowdsourcing, but this remains an expensive and laborious process, and it is unclear which dimensions of narration are captured by the crowdsourced stories. Intuitively, commonsense axioms [Gordon and Hobbs, 2017] can explain stories, yet, grounding stories to these axioms is non-trivial. A cheap way to generate substantial quantities of data for question answering is by leveraging large-scale, freely available knowledge graphs [Ma et al., 2021], but this idea has not been employed to generate stories at scale so far.

In this paper, we investigate the generation of stories at scale, by combining commonsense axioms with commonsense knowledge graphs to produce stories annotated with common sense. The commonsense axioms serve as a well-understood and limited set of primitives that can explain agent behavior and expectations in a given eventative circumstance. Commonsense knowledge graphs guarantee that the content of the story corresponds to interesting phenomena, such as causality or object properties. The generated stories can then be used to enhance the ability of language models to understand stories and explain their decisions. The contributions of this paper are as follows:

1. We investigate the role of commonsense axioms for understanding stories. Our analysis shows that axioms can describe a useful set of phenomena in the ROCStories [Mostafazadeh et al., 2017] story completion corpus, and they can be grounded to knowledge types in modern commonsense knowledge graphs. **(Section 2)**

2. We demonstrate the potential of generating stories by combining axioms with commonsense knowledge graphs. We associate three exemplar story types with their underlying axioms, and extract hundreds of thousands of questions from an existing knowledge graph. **(Section 3)**

3. Our human evaluation reveals that the generated stories are of good quality and that they can be explained by the underlying axioms, thus supporting the overall approach. We also show that our method generates more sensible stories compared to generative language models. **(Section 4)**

4. We discuss future research towards generating better stories, providing more comprehensive explanations, and building models that can explain any story with axioms. **(Section 5)**

## 2. Understanding stories with axioms

### 2.1 Commonsense axioms

Gordon and Hobbs [2017] consolidate theories of commonsense knowledge into a set of axiomatic formalizations, i.e., abstract commonsense relations between concepts in first-order logic, aiming for both high competency and coverage. They devise 1,400 axioms organized into 29 commonsense psychology theories and 16 background theories. A human-readable description of 16 axioms that we use in this paper is provided in Table 1. The axioms facilitate exploration of human-like commonsense reasoning by artificial intelligence researchers,

---

**Substitution (7.1)** Two items, $I$ and $I2$, are substitutes for each other if they play the same role in their corresponding eventualities.

**Change of state to (14.8)** One eventuality $E$ may change into another, inconsistent one, by modifying ($M$) the properties for a common entity $O$. The change is performed by an instrument $I$ at a location of the change, $L$.

**Agent causality (15.6)** The chief property of an agent $A$ is that they, defeasibly, are capable of causing some events.

**Object of (15.9)** $O$ is the object of an event $E$ if $E$ is a change in $O$, or recursively if the event is a causal chain of subevents, the final event of which has $O$ as its object.

**Instrument of (15.10)** $I$ is an instrument of an event $E$ if the agent causes a change in $O$, and that causes $E$ or the end state in $E$.

**At location (18.9)** An external entity $I$ can be at component $C$ within a spatial system $L$, where $C$ is a physical object, and $I$ is unspecified.

**Similar in that (22.1)** Two things, $I$ and $I2$, can be similar with respect to a specific property $G$, allowing them to substitute each other with respect to this property.

**Different in that (22.2)** Two things, $I$ and $I2$, can be different with respect to a specific property $G$, which means that they cannot substitute each other with respect to this property.

**Expectation (26.1)** Agent $A$ at time $T_0$ expects $E$ to happen at time $T_1$.

**Expectation confirmation (26.8)** $A$'s expectation that $E$ will happen at time $T_1$ is confirmed.

**Expectation violation (26.9)** $A$'s expectation that $E$ will happen at time $T_1$ is disconfirmed.

**Achieved goal (28.21)** $G$ is a goal that $A$ has at time $T_0$ and achieves by time $T_1$. A goal that has been achieved is an eventuality that was once a goal and now really exists.

**Unachieved goal (28.22)** $G$ is a goal that $A$ has at time $T_0$ that does not obtain by time $T_1$, i.e., the eventuality $G$ was a goal of $A$'s at time $T_0$ and it does not really exist at time $T_1$.

**Good for (28.37)** Eventuality $E$ is good for agent $A$, as it contributes causally somehow to the achievement of one of its goals, $G$.

**Bad for (28.38)** Eventuality $E$ is bad for agent $A$, as it it contributes causally somehow to the nonachievment of one of its goals, $G$.

**Disappointed (49.67)** An agent $A$ is disappointed that eventuality $E$ occurred. Disappointment results when an event and its negation are both anticipated, the event is good for the person, and the event does not actually occur.

---

Table 1: Axioms from [Gordon and Hobbs, 2017], which are used in our story types.

linguists, and cognitive and social psychologists. The focus is primarily on naive psychology, albeit the complexity of this aspect of human intelligence led to inclusion of additional, non-psychological aspects. For instance, defining agent goals requires axioms on causality and temporality. The axiomatic theories are broadly divided into two groups: 1) fundamental theories, such as eventualities, sets, and defeasibility; 2) content theories, which include high-level areas of abstraction, describing aspects of causality, structure of complex events, and agent goals and plans. In contrast to prior work that has used event sequences, commonsense axioms are focused on agent-based psychology, more comprehensively capturing the motivations, reasoning, and goals of an agent in the context of a narrative.

## 2.2 Axioms in ROCStories

We test the applicability of these axioms by assessing whether they can explain stories from a popular corpus: ROCStories. ROCStories contains 98,162 five-sentence stories for

|           | Unmet expectations | Alternatives |
|-----------|--------------------|--------------|
| **story** | There was this amazing Italian place down the street from Alice. She used to go there almost every week. She loves the pastas that they serve there. One day she tried to go there and found out they closed down. | Ben was excited to go fishing with his grandfather for the first time. However, he found putting the worms on the hook to be disgusting. He accidentally pricked his finger with one of the hooks, too. Ben soon realized that fishing was not for him. |
| **correct** | Alice was upset she had to settle for Chinese food. | Ben asked his grandfather if they could watch a movie instead. |
| **wrong** | She invited all her friends to join her at that restaurant. | Ben asked his grandfather when they could go fishing next. |
| **axioms** | Unexpect, Prefer, Prevent | Anticipate happy, Disgusted, Painful, Substitution |

Table 2: Example stories from ROCStories, with their corresponding axioms.

evaluating story understanding. This corpus has been created by crowdsourcing, and covers a wide range of story domains. Its Cloze task is to select a correct story ending from two possible candidates, and contains 3,744 instances. For our analysis, we randomly sample 50 stories from the ROCStories Cloze task stories, and manually annotate them with applicable axioms that explain why a story ending is (not) expected.

Our analysis shows that the stories can be explained with the CS axioms defined by Gordon and Hobbs [2017]. They mainly involve reasoning about agent goals, plans, and emotions, as well as event sequences. A smaller portion relies on complementary axioms about actions under constraints, changes of an agent's mental state, or modeling of another agent's mental state. The majority of the incorrect endings can be explained by counterfactual reasoning, e.g., having little money is contradictory to having no money at all. Table 2 presents example stories with their corresponding axioms. The story about unmet expectations is explained by axioms about unexpected events, agent preferences, and prevented events due to causality. The story about alternatives is based on understanding about anticipation of positive events, disgust, pain, and substitution of activities.

While axioms can explain various phenomena of storytelling, we note that some gaps need to be filled by commonsense knowledge. For instance, the stories in Table 2 assume understanding of temporality (e.g., a closed down restaurant will likely not re-open soon) and similarity (e.g., both fishing and watching a movie are leisure activities).

## 3. Generating stories with axioms and commonsense knowledge

Inspired by the analysis in the previous section, here we investigate how axioms can be combined with CS knowledge graphs in order to generate a large number of stories. Our method is presented in Figure 1. We consider three story types: unmet expectations, alternatives, and object modifications. We formalize a story type by associating it with underlying axioms and commonsense knowledge relations. We use the formalization to extract paths from a commonsense knowledge graph. The path objects are used to fill a story type template, resulting ultimately in a story in natural language. In addition, the paths, combined with the underlying axioms, are used to generate a human readable

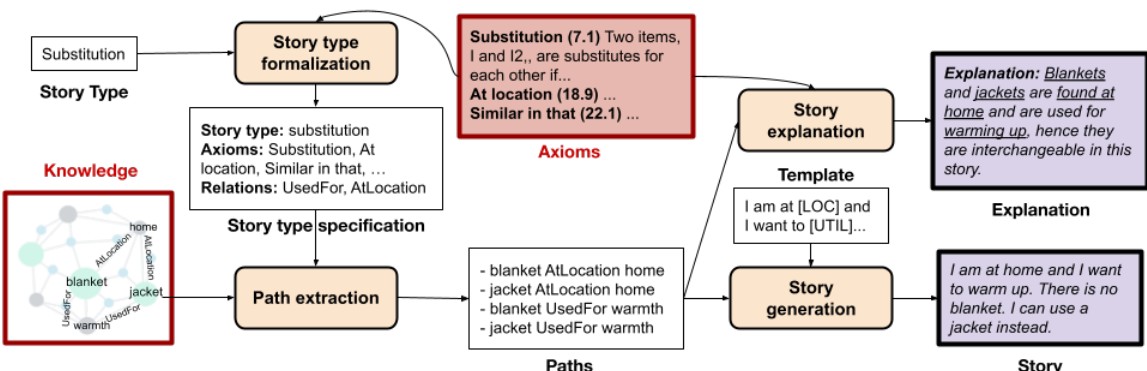

Figure 1: Overview of our story generation method.

explanation. Next, we define and formalize the three story types, we describe the leveraged commonsense knowledge graphs, and we detail our data collection procedure.

### 3.1 Story type formalization

**Type I: Unmet expectations** Situations come with particular expectations. One expects that a refrigerator is located in the kitchen, that it is not empty, and that its contents are colder compared to other items in the room. We formalize a version of unmet expectations, consisting of three elements: a location $L$, a container $C$ that is found in that location, and an item $I$ which is often found in the container $C$. We connect the three elements in a natural language sentence based on a template, namely:

*Q: I went to the (L). There was a (C) but it had no (I). Am I disappointed? A: Y/N*

**Type II: Alternatives** Navigating in the world requires agents to be flexible in terms of their plans, and able to understand the contextual suitability to replace one item with another. For instance, if one wants to eat and is unable to find a restaurant nearby, a café is likely to be a suitable substitute. We formalize stories about alternatives with four elements: a location $L$, a goal $G$, an item $I$, and its potential alternative $I2$. We connect the four elements in a natural language sentence based on a template, namely:

*Q: I went to (L) and I wanted to (G). There was no (I), can I use (I2) instead? A: Y/N*

**Type III: Object modifications** Understanding the typical effect of one object on another is another key aspect of grasping situations. A coffee in a microwave is likely to get warmed up, popcorn grow in a microwave, while a refrigerator will get both the coffee and the popcorn cold. We define four elements in the object modification scenario: a location $L$, an item $I$, an affected object $O$, and state modification $M$. We use the following template:

*Q: There is (I) in the (L). Can it (M) the (O)? A: Y/N*

### 3.2 Commonsense knowledge graphs

The structured sources of commonsense knowledge can be categorized into: commonsense knowledge graphs, common knowledge graphs, visual sources, and lexical sources [Ilievski et al., 2021a]. These sources typically represent knowledge in the form of a directed triple $< h, r, t >$, consisting of a head $(h)$, relation $(r)$, and a tail $(t)$. We instantiate candidate

|  | Unmet expectations | Alternatives | Object modifications |
|---|---|---|---|
| **Axioms** | 6-At location | 1-Substitution | 2-Change of state to |
|  | 9-Expectation | 6-At location | 3-Agent causality |
|  | 10-Expectation confirmation | 7-Similar in that | 4-Object of |
|  | 11-Expectation violation | 8-Different in that | 5-Instrument of |
|  | 12-Achieved goal |  | 6-At location |
|  | 13-Unachieved goal |  |  |
|  | 14-Good for |  |  |
|  | 15-Bad for |  |  |
|  | 16-Disappointed |  |  |
| **Paths** | I AtLocation C | I UsedFor G | I AtLocation L |
|  | C AtLocation L | I2 UsedFor G | I UsedFor (M,O) or |
|  |  | I AtLocation L | I CapableOf (M,O) |
|  |  | I2 AtLocation L |  |

Table 3: Paths for the three story types. *Unmet expectations:* I went to the (L). There was a (C) but it had no (I). Am I disappointed?; *Alternatives:* I went to (L) and I wanted to (G). There was no (I), can I use (I2) instead?; *Object modifications:* There is (I) in the (L). Can it (M) the (O)?

stories by path traversal over the Commonsense Knowledge Graph (CSKG) [Ilievski et al., 2021b], which consolidates seven such sources, including ConceptNet [Speer et al., 2017], ATOMIC [Sap et al., 2019a], and Visual Genome [Krishna et al., 2017]. At present, for each story type, we manually define its paths over CSKG in accordance with relevant axioms. We envision these to be automatically generated by a trained language model in the future.

### 3.3 Data collection and filtering

For each story type, we collect candidate stories by traversing CSKG. The designed paths for each of the story types are presented in Table 3. Here, we also outline the axioms that collectively explain each story. Following Gordon and Hobbs [2017], we provide a formal definition of each axiom in Table 1. The candidate stories are further filtered in accordance to the relevant axioms. We next detail the procedure for each story type.

**Type I: Unmet expectations** Stories about *unmet expectations* are based on agent expectations and goals, their realization in a given location, and their associated sentiment. Specifically, if an agent expects that an item $I$, which is good for them, is found in a container $C$ in a location $L$, then not finding this item is disappointing for the agent. Conversely, if an item $I$ is not expected in a container $C$ in a location $L$, then not finding this item does not disappoint the agent. The initial number of sentences generated from CSKG is 121,148. First, we filter out candidates where $L$ is not a real location, based on a frequency-based heuristic: we only keep the top-100 most frequent locations. This reduces the number of sentences to 49,966. Next, we remove sentences whose item is an animate object. We do this by constructing a WordNet-based list of terms which are subclasses of people and animals.[1] To connect the disappointment of an agent to the "goodness" of the

---

1. Synsets: animal.n.01, person.n.01, people.n.01, peoples.n.01

outcome, we filter the remaining sentences based on sentiment analysis. If both the item and its container have a positive sentiment, then we keep this story and infer that not finding such an item is disappointing (answer 'yes'). To balance the data, we generate as many additional sentences whose item is unexpected at a given location, which does not lead to disappointment (answer 'no'). This procedure results in 80,888 stories, whose answers are split equally between 'yes' and 'no'.

**Type II: Alternatives** Stories on *alternatives* rely on axioms about substitution, location of objects, and similarity/distinctness. Namely, two items that are used for the same purpose and coincide in their location, may serve as alternatives to each other. The initial number of sentences with this method is 14,949, all of which have an answer 'Yes'. We use a frequency-based heuristic to select locations, and only keep the 100 most frequent locations. After this step, we have 9,733 stories left. Next, we avoid alternatives which are synonymous (e.g., TV and television; dentist office and dental office). We filter these out in two ways: 1) by generating a list of synonyms based on WordNet, and filtering sentences where the item and its alternative are synonyms; and 2) by the Levenshtein and Jaccard string similarity metrics, by ensuring that their Levenshtein similarity is higher than 0.9 and their Jaccard similarity is below 0.6. The final set of sentences with answer 'yes' after this filtering step is 3,772. In order to generate negative samples, for each existing story we generate an alternative story with an answer 'no', where the alternative item is randomly sampled from the set of alternatives to the other stories which are not correct alternatives to the current one according to CSKG. This results in a total set of 7,544 stories.

**Type III: Object modifications** Stories about *object modifications* rely on understanding about locations, as well as the instrument and the object of a change of state. Namely, an instrument causes a specific change of state to an object, at a given location. From CSKG, we obtain instrument utilities as compounds of modification $M$ and object $O$ (e.g., bake a cake). We split these utilities into the constituent object $O$ and modification $M$, which requires that the utility must be a compound of a verb and a noun. The initial set of stories generated by this method is 214,878. The top-100 most frequent locations cover 69,509 stories. After discarding animate objects, our process generates 17,486 stories, with a large variety of answers. We generate stories with a negative answer, by randomly sampling an action that does not apply to a given object. We end up with 29,392 stories, whose answers are split equally between 'yes' and 'no'.

## 4. Evaluation

Our method resulted in 118K short stories: 81K about unmet expectations, 8K about alternatives, and 29K about object modifications. Each story is associated with a binary question, whose answers are equally balanced between the two categories ('yes' and 'no'). Next, we report on two experiments that measure the quality of the generated stories. We evaluate the accuracy of the stories by measuring agreement between our method and humans. We compare the sensibility and interestingness of our stories to alternative stories generated by a sequence-to-sequence language model.

| Prompt | GPT-2 | Our method |
|---|---|---|
| I went to the street corner. | I went to the store but never saw someone who had a camera there. | ***There was a mailbox but it had no letter slot. (S, I)*** |
| There was clock in the table. | *'There is a timer ticking, we are about to leave.' (I)* | **I wanted to use it to keep a(n) time. (S)** |
| I went to the city. | ***"It's my first time," she said. (S,I)*** | There was a runway but it had no plane. |

Table 4: Stories generated by our method and by GPT-2. The more sensible story is marked in bold. The more interesting story is marked in italic.

### 4.1 Measuring agreement with humans

**Human evaluation** We randomly select 450 stories (150 from each story type), and ask crowd workers to answer them. The crowd workers can choose one of the following three answers: "yes", "no", and "I don't know". We obtain 3 judgments per story, resulting in 1,350 judgments in total. Among these, 9.55% (129/1,350) of the judgments are "I don't know". We pick the majority vote for each question, by comparing the number of "yes" and "no" judgments. This results in 207 answers "yes", 208 "no", and 35 ties.

We next compare the majority answer according to the crowd, to the answer according to our story generation procedure. The humans agree with our method on 71.5% (322 out of 450) of all the answers, or 77.6% (322/415) if we discount the ties. We also compute agreement per story type. The agreement is lowest on stories about unmet expectations (52.67%, or 60.3% without ties), and highest on stories about alternatives (86% and 87.75%, respectively). The agreement on the stories about object modifications is 76% (83.21% without ties). The set of 322 validated stories can be found online at: `shorturl.at/fivyA`.

Our qualitative analysis identifies three common causes for disagreement: 1) **Incorrect negative sampling:** in some cases, we sample a generic, yet viable, location as a negative sample, which results in a false negative. Such a case is "I went to the michigan. There was a theater but it had no theater ticket. Am I disappointed?", to which the crowd workers rightfully answer "yes". 2) **Missing agent goals:** Most of the disagreements are observed on stories about unmet expectations. These stories currently lack information about the agent goal. For instance, not finding a date book on a desk is only disappointing if the agent needs it. 3) **Remaining ambiguity:** Some stories are imprecise and ambiguous, which allows for different interpretations, and consequently, different answers. Such a case is the story "There was yard in the house. Can it be used to measure a(n) distance?", to which the crowd says "no", while our method answers with "yes".

### 4.2 Comparison to a language model generator

In Section 1 we claimed that language models cannot generate high-quality stories, and we expect that our commonsense-based method is able to improve on this task. Here we test these claims, by comparing the quality of the stories generated by our method to those generated by a state-of-the-art sequence-to-sequence model, GPT-2 [Radford et al., 2019]. We randomly pick 100 of our generated stories, and change them into affirmative sentences (e.g., 'can it be used' becomes 'I decided to use it'). Then, we leave out the story ending sentence, and use GPT-2 to provide an alternative completion. We ask crowd workers to

choose between our story and the one by GPT-2, based on two questions: 1) which story makes more sense; 2) which story is more interesting. We collect three judgments per story.

The results confirm our hypothesis. In 73% of the cases, our stories are judged to be more sensible than the ones generated by GPT-2 (as opposed to 27% of the cases where GPT-2 generated a more reasonable story). As it can be expected, the interestingness of the stories is much more similar: 52% votes for our method, and 48% for GPT-2. These findings demonstrate that our method can generate more reasonable and well-rounded stories than GPT-2, whereas both methods produce similarly interesting stories. This finding is illustrated in Table 4 with exemplar stories and their corresponding judgments.

## 5. Discussion

**How to generate better stories?** Our method described in this paper was able to yield over a hundred thousand stories that correspond to three types, and can be easily scaled up to more story types or larger knowledge graphs. Our human evaluation showed that the generated stories are moderately accurate. They are more sensible than, yet similarly interesting to, those generated by a language model. More interesting stories could be generated by more complex graph patterns and a richer set of axioms, e.g., including causality and agent goals. Yet, given the noise in the graph, it is to be expected that more interesting stories will be produced at the expense of their soundness. Thus, generation of stories which are both more sound and interesting depends on the quality of the available knowledge in CSKG. Generating better stories requires further consolidation of CSKG, to address reported issues of ambiguity, variance, and incompleteness [Ilievski et al., 2021a].

**How to provide a more comprehensive explanation?** Explaining a story requires understanding of the relevant axioms, but also possession of commonsense knowledge. For example, in order to understand that two girls becoming friends may cause one of the girls to be happy, we need to understand both the key axioms (e.g., happiness is a consequence of achieving a goal) as well as universal human knowledge (e.g., people generally have an intrinsic goal to make friends). A crucial next step for our research is to enhance the story explanations, by filling gaps in the axioms with relevant commonsense knowledge.

**How to build a model that can explain any story?** In Section 2.2, we showed that the commonsense axioms in [Gordon and Hobbs, 2017] cover all stories which we analyzed manually. An explanatory language model, like WT5 [Narang et al., 2020], can thus be trained to explain any story in a corpus like ROCStories. Generating training data for explaining stories with commonsense axioms would require the axioms to be grounded to their corresponding stories. This could be performed in a semi-automatic manner: an automatic method would propose possible axioms, e.g., by distant supervision based on a small set of manually annotated stories, such as those in Section 2.2. In a subsequent step, humans (e.g., crowd workers) would validate the axioms with a lowest estimated confidence.

## 6. Related work

Two major research directions relate to our paper: 1) story generation and understanding, and 2) benchmark generation with commonsense knowledge graphs.

### 6.1 Story generation and understanding

ROCStories [Mostafazadeh et al., 2017] is a corpus and task for commonsense story completion: given four sentences, models are asked to complete the story with the final sentence. LIGHT [Urbanek et al., 2019] is a framework for assessing models' ability to generate natural dialogue, actions, and emotes in hundreds of crowdsourced environments. Fan et al. [2020] devises a machine learning approach that learns to create worlds based on the adventure environments collected in LIGHT. Embodied QA [Das et al., 2018] takes this a step further, by placing an embodied agent in a generated world environment. Rather than generating a small number of expressive and high-quality worlds, our focus is on generating a large number of stories dynamically from freely available commonsense knowledge graphs based on well-understood axioms of human cognition.

Prior work has also attempted to generate stories at scale. One idea is to divide the generative task into two phases: event sequence generation and transformation of these events into English sentences [Ammanabrolu et al., 2019, Martin et al., 2018, Yao et al., 2019]. Radford et al. [2018] show that generative language models can generate better story endings upon minimal adaptation of their input. Language models can be adapted to perform controllable language generation, by being combined with user-specified bag of words [Dathathri et al., 2020], ending valence and keywords [Peng et al., 2018], or homophones [He et al., 2019]. The template-based story generation model by Chaturvedi et al. [2017] uses the sequence of events described in the story, the evolution of sentiment and emotional trajectories, and topical consistency. See et al. [2019] control four attributes of multi-turn conversations (repetition, specificity, response-relatedness, and question-asking) and measure their impact on human judgments of quality. Recognizing that commonsense knowledge is a key aspect of story comprehension, Rashkin et al. [2018] provide an annotation formalism for labeling of the mental states of the characters in short commonsense stories, whereas the story ending selection model by Chen et al. [2019] combines narrative sequences, sentiment evolution, and commonsense knowledge. In the latter work, commonsense knowledge is included based on ConceptNet's numberbatch embeddings [Speer et al., 2017]. Our work complements such notable efforts on story generation, because: (1) we use common sense as the underlying framework, based on axiomatic understanding coupled with large-scale common knowledge, and (b) our framework has the ability to provide supervision in the form of commonsense reasoning.

### 6.2 Generating benchmarks from commonsense knowledge graphs

Pattern-based extraction from ConceptNet [Speer et al., 2017] has been used to create commonsense benchmarks for multiple-choice question answering [Talmor et al., 2018] and logical probing [Zhou et al., 2020]. Parts of ATOMIC [Sap et al., 2019a] have been leveraged in the construction of the SocialIQA [Sap et al., 2019b] dataset. Besides for benchmarking, commonsense knowledge has also been employed to enhance the ability of language models to perform zero-shot reasoning across tasks. An efficient way to generate substantial data for question answering is by leveraging large-scale, freely available commonsense knowledge graphs [Ma et al., 2021]. In this paper, we extend this idea by combining commonsense knowledge with cognitive axioms in order to generate stories at scale.

## 7. Conclusions

This paper proposed a method that combines commonsense axioms with large knowledge graphs in order to understand and generate stories at scale. We showed that axioms can describe a useful set of phenomena in a popular corpus with 100K stories, ROCStories, and they can be grounded to knowledge types in modern commonsense knowledge graphs. We generated hundreds of thousands of stories with questions for three exemplar story types, and we evaluated their sensibility and interestingness through crowdsourcing. Our human evaluation showed that are stories are largely accurate, and that they are more sensible than those of a language model, yet of similar interestingness. Based on these findings, we discussed three open challenges for generating and explaining stories with common sense as the underlying framework: creating better stories, providing more comprehensive explanations, and building models that can explain any story with commonsense axioms.

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
