# OpenReview forum: "Story Generation with Commonsense Knowledge Graphs and Axioms"
_AKBC.ws/2021/Workshop/CSKB — CSKB_

### Official Review · Reviewer_hd8h · 2021-09-13
**Story Generation with Commonsense Knowledge Graphs and Axioms**

**Rating:** 7
**Confidence:** 4

**Review:**

The paper explores automatically generating stories at scale using common sense axioms and commonsense knowledge graphs. They first demonstrate through manual analysis that in fact commonsense axioms can be sufficiently used to explain why a story ending is correct or not in a popular Story Cloze test task. Three story types are investigated (aligned with CS axioms) for which they created binary question templates to connect the elements. The story candidates are then generated by manually defined path traversal (for each story type) over multiple CSKG, namely ConceptNet, ATOMIC, and Visual Genome.
They conducted human evaluation for assessing the quality of the generated stories using their proposed approach.

Strengths:

* A good job on connecting commonsense axioms and its applicability to story understanding and generation tasks.
* Generating commonsense guided stories at scale
* Nice discussion section + future direction on story explanation

Weaknesses:
* Some analyses lack enough supporting evidence or qualitative examples.
* It is not clear how the approach can be extended for more types of story formalization (other than the three discussed in the paper). In other words, the approach might not be as effective when it comes to generating a diverse set of stories with various styles.

Comments:

* Table 3 is better to be placed earlier in the paper.

* The ROCStories has 3,744 Story Cloze Test instances where the task is to select correct story ending and not 98,162 which is the total number of stories in ROCStories corpus.

* The analysis in Section 2.2 could have been more helpful but sadly the author did not provide enough statistics or results. What percentage of the 50 stories could be explained by CS axioms? What is the portion for each type of axiom? An expanded discussion on this section is very helpful and can shed lights on an interesting direction of using your approach for providing automatic story explanations (some points are made in the discussion section which is nice.)

* How are the story types defined? Are they defined based on some manual analysis of ROCStories corpus, similar to the section 2.2? The author could have expanded on the motivation behind selecting a limited number of story types (3).

* It is not clear whether the goal of the paper is to generate the full story or the story ending? While the author claimed a story generation task, the experiments focused on generating a single next plausible sentence and not a full story. It would be nice to make this clear by providing the example of stories generated by your method. How many sentences they have on average? Examples in Table 4 are ending sentences.

* It might not be very meaningful for an item or container to have sentiment polarities as discussed in the paper for unmet expectation. Can this be the reason for the lowest human agreement on this type of story?

* Comparing stories generated by GPT2 stories vs. your mode in terms of interestingness/plausibility is not a fair comparison as GPT2 is not controlled to generate in such settings. A better comparison should be made on the full story and not on the immediate next sentence only.

* The statement about the best performing model accuracy on Story Cloze task (75%) should be updated according to the advances in recent PTLMs. The SOTA score is significantly higher now.

---

### Decision · Program_Chairs · 2021-09-18

Accept